# Next-generation unnatural monosaccharides reveal that ESRRB O-GlcNAcylation regulates pluripotency of mouse embryonic stem cells

Yi Hao[1,2,6], Xinqi Fan[1,3,6], Yujie Shi[1,3], Che Zhang[1,3], De-en Sun[1,3], Ke Qin[1,3], Wei Qin[1,2], Wen Zhou[1,3] & Xing Chen [1,2,3,4,5]

Unnatural monosaccharides such as azidosugars that can be metabolically incorporated into cellular glycans are currently used as a major tool for glycan imaging and glycoproteomic profiling. As a common practice to enhance membrane permeability and cellular uptake, the unnatural sugars are per-O-acetylated, which, however, can induce a long-overlooked side reaction, non-enzymatic S-glycosylation. Herein, we develop 1,3-di-esterified N-azidoacetylgalactosamine (GalNAz) as next-generation chemical reporters for metabolic glycan labeling. Both 1,3-di-O-acetylated GalNAz (1,3-Ac$_2$GalNAz) and 1,3-di-O-propionylated GalNAz (1,3-Pr$_2$GalNAz) exhibit high efficiency for labeling protein O-GlcNAcylation with no artificial S-glycosylation. Applying 1,3-Pr$_2$GalNAz in mouse embryonic stem cells (mESCs), we identify ESRRB, a critical transcription factor for pluripotency, as an O-GlcNAcylated protein. We show that ESRRB O-GlcNAcylation is important for mESC self-renewal and pluripotency. Mechanistically, ESRRB is O-GlcNAcylated by O-GlcNAc transferase at serine 25, which stabilizes ESRRB, promotes its transcription activity and facilitates its interactions with two master pluripotency regulators, OCT4 and NANOG.

[1] College of Chemistry and Molecular Engineering, Peking University, Beijing, China. [2] Peking-Tsinghua Center for Life Sciences, Peking University, Beijing, China. [3] Beijing National Laboratory for Molecular Sciences, Peking University, Beijing, China. [4] Synthetic and Functional Biomolecules Center, Peking University, Beijing, China. [5] Key Laboratory of Bioorganic Chemistry and Molecular Engineering of Ministry of Education, Peking University, Beijing, China. [6]These authors contributed equally: Yi Hao, Xinqi Fan. Correspondence and requests for materials should be addressed to X.C. (email: xingchen@pku.edu.cn)

Since the development of metabolic glycan labeling (MGL) in late 90s, it has emerged as a widely used method for tagging glycans in live cells with fluorophores for imaging or enrichment probes for glycoproteomic profiling[1–3]. Unnatural monosaccharides or unnatural sugars—analogues of monosaccharides containing a bioorthogonal functional group (e.g., the azide or alkyne)—are administered to the culture medium, taken up by the cells, in which they are accepted by the glycan biosynthetic enzymes and thus metabolically incorporated into cellular glycans. Subsequent bioorthogonal reactions, such as Cu(I)-catalyzed azide-alkyne cycloaddition (CuAAC or click chemistry)[4,5], are performed to selectively conjugate the labeled glycans with desired probes. A critical design principle for MGL is the tolerance of the glycan biosynthetic pathways for respective unnatural monosaccharides, which now include azido or/and alkynyl analogues of N-acetylmannosamine (ManNAc), fucose, N-acetylglucosamine (GlcNAc), and N-acetylgalactosamine (GalNAc). Notably, these unnatural monosaccharides are usually administered in a per-O-acetylated form to improve membrane permeability and cellular uptake. Once inside the cells, the per-O-acetylated unnatural sugars are presumably hydrolyzed by cytosolic esterases, releasing the deprotected unnatural sugars for metabolic incorporation[6,7].

Recently, we discovered that most of the per-O-acetylated monosaccharides and their azido/alkynyl analogues can non-specifically react with cysteine (Cys) residues of various intracellular proteins through a non-enzymatic S-glycosylation reaction[8]. This artificial chemical reaction, which has been overlooked for two decades, may interfere with glycoproteomic profiling, for example, causing false positives in proteomic identification of O-linked GlcNAc modified proteins. Dynamically regulated by a pair of enzymes, O-GlcNAc transferase (OGT) and O-GlcNAcase (OGA), O-GlcNAcylation modifies specific serine or threonine residues of various intracellular proteins with a β-O-linked GlcNAc monosaccharide, which regulates many important biological processes, such as transcription and stress response[9,10]. Per-O-acetylated GalNAz (Ac4GalNAz) and per-O-acetylated 6-azido-6-deoxy-GlcNAz (Ac36AzGlcNc) are two currently used O-GlcNAc reporters;[11,12] they both can non-specifically S-glycosylate various intracellular proteins, including some of the identified O-GlcNAcylated proteins[8]. It should be noted that different per-O-acetylated monosaccharide analogues have varied S-glycosylation reactivity. For example, Ac36AzGlcNAc showed much weaker S-glycosylation-induced labeling in cell lysates than several other ones, such as Ac4GalNAz[8]. Accordingly, siRNA knockdown of OGT in Ac36AzGlcNAc-treated cells resulted in the loss of most of the labeling[13]. Nevertheless, even slight non-specific S-glycosylation may interfere with proteomic identification of O-GlcNAcylated proteins, particularly when the modification sites are not identified. Although naked or unprotected monosaccharide analogues, such as N-azidoacetylgalactosamine (GalNAz), can avoid S-glycosylation, they need to be administered using high concentrations in the millimolar range[7,8].

Here we report the development of next-generation unnatural monosaccharides, which can be taken up by cells with high efficiency and more importantly, do not induce artificial S-glycosylation. Using GalNAz as a testing case, a series of partially esterified GalNAz, with varied numbers of hydroxyl groups protected by acetate or propionate, are synthesized and evaluated. Among them, 1,3-di-O-propionyl-N-azidoacetylgalactosamine (1,3-Pr2GalNAz) exhibits the highest labeling efficiency and specificity. Applying 1,3-Pr2GalNAz for probing O-GlcNAcylation of pluripotency transcription factors in mouse embryonic stem cells (mESCs), we discover that ESRRB, a critical transcription factor for establishing self-renewal and pluripotency[14], is O-GlcNAcylated at Ser 25. ESRRB O-GlcNAcylation enhances the protein stability and binding of ESRRB to OCT4 and NANOG, two master pluripotency regulators, is augmented by O-GlcNAcylation. Moreover, O-GlcNAcylation on ESRRB promotes its transcriptional activity. Ablation of ESRRB O-GlcNAc impairs its function for maintaining mESC self-renewal and pluripotency, both in cell culture and during teratoma formation in mice. The next-generation unnatural sugars developed in this work therefore provide an improved MGL platform for probing O-GlcNAc biology.

## Results

**Design and synthesis of partially protected GalNAz.** Considering that per-O-acetylated GalNAz (Ac4GalNAz) can non-enzymatically react with Cys but GalNAz does not have this non-specific S-glycosylation[8], we wondered whether the four O-acetyl groups contributed differentially to S-glycosylation. If this was the case, an ideal unnatural sugar would be partially protected, having the acetyl group(s) causing S-glycosylation removed, so that cell membrane permeability can be improved and simultaneously artificial S-glycosylation avoided (Fig. 1a).

To test our design, we chemically synthesized 3,4,6-tri-O-acetyl-N-azidoacetylgalactosamine (3,4,6-Ac3GalNAz; compound 1)[15], 1,3,6-tri-O-acetyl-N-azidoacetylgalactosamine (1,3,6-Ac3GalNAz; compound 2), 1,3-di-O-acetyl-N-azidoacetylgalactosamine (1,3-Ac2GalNAz; compound 3), and 1,3-di-O-propionyl-N-azidoacetylgalactosamine (1,3-Pr2GalNAz; compound 4) with an overall yield of 78%, 17.9%, 26.9% and 29.1%, respectively (Fig. 1b and Supplementary Note 1). Of note, the propionyl group with a longer alkyl chain was employed for 4 to further increase the hydrophobicity and enhance the membrane permeability. The acyl migration from the 3-hydroxyl group to the 4- and 6-hydroxyl groups for 3 and 4 was monitored in $D_2O$ by NMR spectroscopy (Supplementary Fig. 1). Slight acetyl migration was observed for 3 starting from 4 h. No migration was detected for 4 for up to 48 h, in agreement with previous studies showing bulky acyl groups have less tendency to migrate[16,17].

**High efficiency and specificity of 1,3-di-esterified GalNAz.** To assay the artificial S-glycosylation, HeLa cell lysates were incubated with individual unnatural monosaccharides. The direct and non-enzymatic reaction of azido sugars with proteins were detected by click-labeling with alkyne-Cy3, followed by in-gel fluorescence scanning (Fig. 2a–d and Supplementary Fig. 2). As two controls, Ac4GalNAz showed Cy3 labeling on various proteins, while GalNAz resulted in negligible labeling. With the 1-O-acetyl group removed, 1 exhibited much higher reactivity to HeLa cell lysates than Ac4GalNAz, indicating that deacetylation at the anomeric oxygen promoted S-glycosylation (Fig. 2a). By changing three acetyl groups onto the 1-, 3-, and 6-hydroxyl groups, 2 showed much lower, but detectable, labeling in cell lysates (Fig. 2b). The labeling of 1, 2, and Ac4GalNAz was completely abolished by pre-treating the lysates with iodoacetamide to block cysteine residues, confirming the occurrence of S-glycosylation on Cys residues (Supplementary Fig. 3). By reducing one more acetyl group while keeping the 1-hydroxyl group protected, 3 showed no S-glycosylation in cell lysates (Fig. 2c). Importantly, with the 1- and 3-hydroxyl groups protected with the bigger propionyl group, 4 did not react with HeLa cell lysates (Fig. 2c). Similar results were obtained for lysates of HEK293T cells (Supplementary Fig. 4). Furthermore, neither 3 nor 4 reacted with purified proteins that are reactive with Ac4GalNAz and 1 (Fig. 2d and Supplementary Fig. 5). In addition, 5 mM glutathione did not react with all the azidosugars, except that the most reactive 1 showed slight reactivity with glutathione (Supplementary Fig. 6). Given that the intracellular concentration of glutathione is

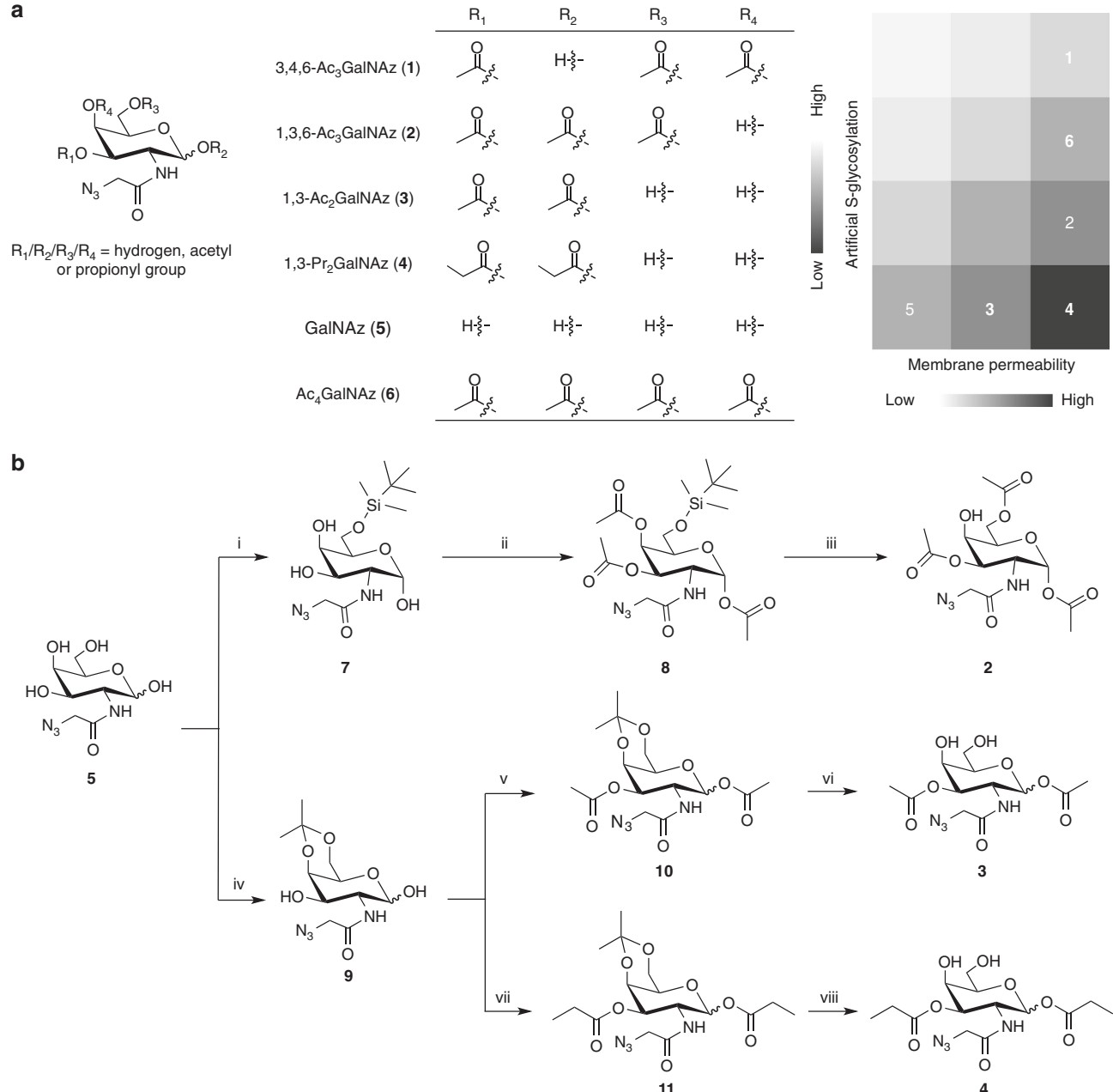

**Fig. 1** Design and synthesis of partially esterified GalNAz. **a** Partially protected GalNAz with varied numbers of acetate or propionate. The greyscale chart shows the membrane permeability and ability of inducing artificial S-glycosylation of the unnatural monosaccharides. **b** Chemical synthesis of **2**, **3** and **4**. i. tert-butyldimethylsilyl chloride, pyridine, room temperature (R.T.), 20 h, 50%. ii. acetic anhydride, pyridine, 0 °C-R.T., overnight, 92%. iii. trifluoroacetic acid, dichloromethane, 0 °C-R.T., 1 h, 39%. iv. 2,2-dimethoxypropane, (±)-camphorsulfonic acid, acetone, 0–4 °C, 1 h, 48%. v. acetic anhydride, pyridine, 0 °C-R.T., overnight, 80%. vi. trifluoroacetic acid, $H_2O$, $CH_3CN$, 0 °C-R.T., 1 h, 70%. vii. propionic anhydride, pyridine, 0 °C-R.T., overnight, 82%. viii. trifluoroacetic acid, $H_2O$, $CH_3CN$, 0 °C-R.T., 1 h, 74%

~1–10 mM[18], the Cys S-glycosylation probably depends on the protein microenvironment surrounding the reactive Cys residues. Taken together, these results demonstrate that 1,3-diesterified GalNAz, **3** and **4**, do not induce non-specific S-glycosylation. They were referred to as $Ac_2GalNAz$ and $Pr_2GalNAz$ in the following of this work.

We next evaluated the metabolic labeling efficiency of $Ac_2GalNAz$ and $Pr_2GalNAz$. HeLa cells were cultured with $Ac_2GalNAz$, $Pr_2GalNAz$, or GalNAz at varied concentrations for 48 h, followed by click reaction with alkyne-Cy5 and analysis of the cell lysates by in-gel fluorescence scanning. Both $Ac_2GalNAz$ and $Pr_2GalNAz$ showed higher incorporation efficiency than

GalNAz (Fig. 2e). Metabolic labeling of HeLa cells with 1 mM GalNAz could achieve reasonably efficient identification of O-GlcNAcylated proteins and modification sites[8]. Using this as a reference, 200 μM $Ac_2GalNAz$ exhibited comparable labeling, and $Pr_2GalNAz$ at the concentration as low as 20 μM achieved the same labeling intensity (Supplementary Fig. 7a). Increasing the concentration of $Pr_2GalNAz$ resulted in significantly higher incorporation in a concentration-dependent manner in the range of 50–500 μM. Moreover, the metabolic incorporation of $Pr_2GalNAz$ was incubation time-dependent (Supplementary Fig. 7b). In addition, comparison of $Pr_2GalNAz$ with GalNAz was performed in a variety of cell lines including HEK293T,

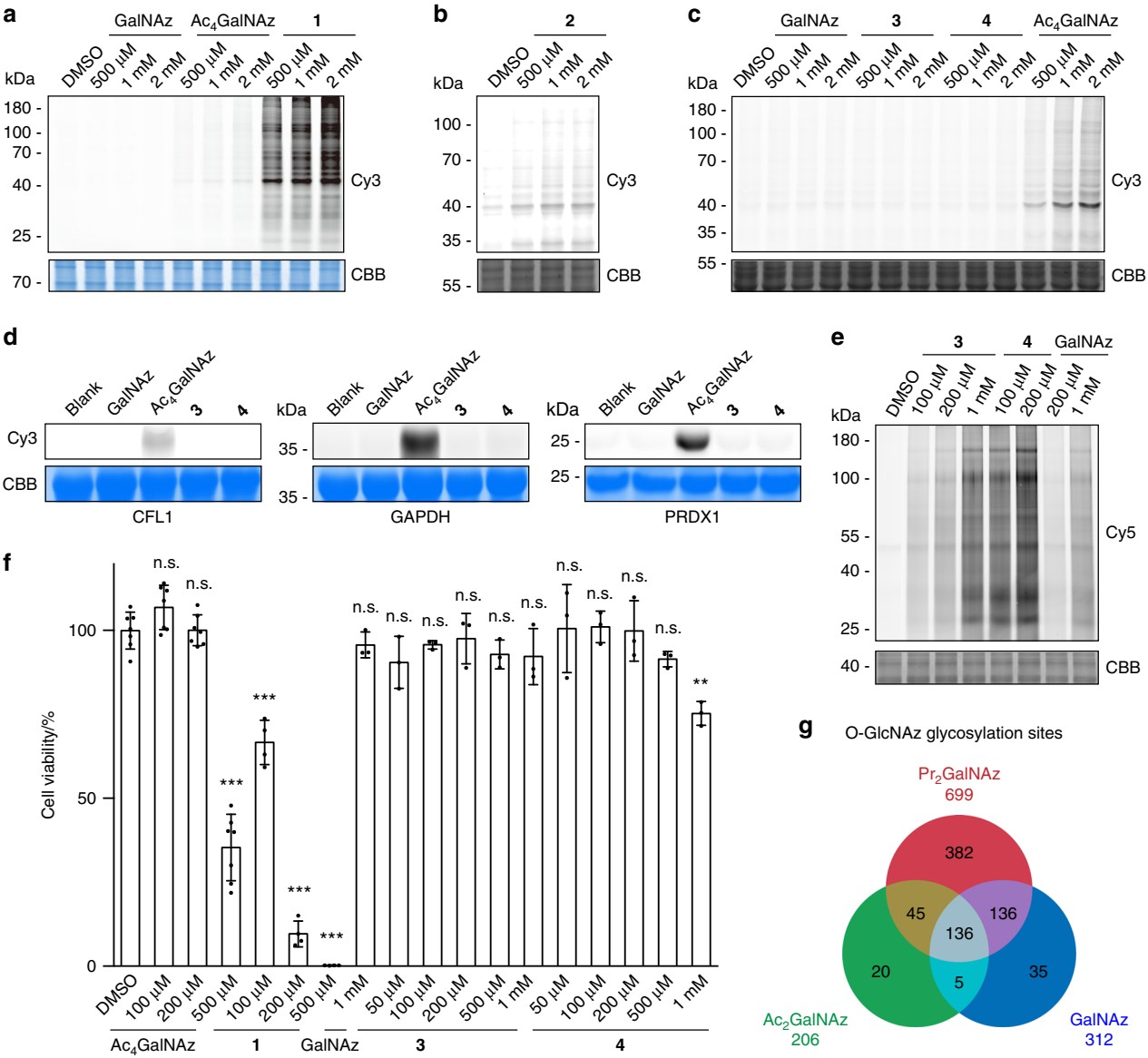

**Fig. 2** Labeling efficiency and specificity of 1,3-Ac2GalNAz and 1,3-Pr2GalNAz. **a–c** In-gel fluorescence scanning showing HeLa cell lysates treated with respective unnatural monosaccharides at varied concentrations for 2 h, followed by reaction with alkyne-Cy3. **d** In-gel fluorescence scanning showing purified CFL1, GAPDH, and PRDX1 treated with 1 mM of respective unnatural monosaccharides for 2 h, followed by reaction with alkyne-Cy3. **e** In-gel fluorescence scanning showing HeLa cells incubated with respective unnatural monosaccharides at varied concentrations for 48 h, followed by reaction with alkyne-Cy5. Coomassie Brilliant Blue (CBB)-stained gels in **a–e** demonstrate equal loading. In **a–e** representative results are from three independent experiments. **f** Cell counting assay by CCK-8 kit showing viability of HeLa cells incubated with respective unnatural monosaccharides at varied concentrations for 48 h. Error bars represent mean ± s.d. Results are from at least three independent experiments (for DMSO and Ac4GalNAz, $n = 7$; for **1**, n = 4; for GalNAz, **3** and **4**, $n = 3$). **P < 0.01, ***P < 0.001, n.s., not significant (one-way ANOVA). **g** Overlap of O-GlcNAz sites identified in HeLa cells treated with GalNAz (based on the data set from ref. [8]), 1,3-Ac2GalNAz and 1,3-Pr2GalNAz. Source data for figures **a–f** are provided as a Source Data file

CHO, Neuro-2a, HT1080, SH-SY5Y, NCI-H1299, A549, NIH/3T3, and MCF-7 cells (Supplementary Fig. 8). In all cell lines tested, Pr2GalNAz showed improved labeling efficiency, with the fold of improvement varied in a cell type-dependent manner.

In the cells, GalNAz enters the GalNAz salvage pathway and forms UDP-GalNAz, which is interconverted with UDP-GlcNAz[11]. Therefore, Ac2GalNAz, Pr2GalNAz, and GalNAz should label both O-GlcNAc and cell-surface glycans. Inhibition of OGT by peracetylated 2-acetamido-2-deoxy-5-thio-D-gluco-pyranose (Ac4SGlcNAc)[19] in HeLa cells treated with Ac2Gal-NAz, Pr2GalNAz, or GalNAz resulted in partial loss of the azide

labeling, supporting that three unnatural sugars can label both O-GlcNAc and cell-surface glycans (Supplementary Fig. 9).

**Cytotoxicity of unnatural monosaccharides.** The cytotoxicity of the five azido sugars was evaluated by treating HeLa cells at varied concentrations for 48 h (Fig. 2f). No significant cytotoxicity was observed for Ac4GalNAz at concentrations up to 200 μM, while 500 μM Ac4GalNAz caused a dramatic loss of cell viability. In accordance with the S-glycosylation reactivity, **1** exhibited much more severe cytotoxicity. For GalNAz, Ac2GalNAz, and Pr2Gal-NAz, no or minimal cytotoxicity was observed at concentrations

up to 1 mM. Similar results were observed in various cell lines, including HEK293T, CHO, Neuro-2a, HT1080, SH-SY5Y, NCI-H1299, A549, NIH/3T3, and MCF-7 cells (Supplementary Fig. 10). These results demonstrate that the newly developed unnatural monosaccharides, Ac$_2$GalNAz and Pr$_2$GalNAz, are not cytotoxic. In addition, the cytotoxicity of unnatural monosaccharides appears to correlate with the artificial S-glycosylation.

**Specific and large-scale identification of O-GlcNAc sites**. Using the chemoproteomic protocol for O-GlcNAcylation profiling[8,20], we performed large-scale identification of O-GlcNAc sites in HeLa cells treated with Ac$_2$GalNAz or Pr$_2$GalNAz (Supplementary Fig. 11). By using an alkyne-biotin tag containing an acid-cleavable linker (alkyne-AC-biotin), the azide-modified peptides were enriched and subjected to analysis by liquid chromatography tandem MS (LC-MS/MS) with electron transfer dissociation (ETD) fragmentation. In all, 206 O-GlcNAc sites and four S-glycosylation sites were identified in HeLa cells treated with 200 μM Ac$_2$GalNAz (Fig. 2g, Supplementary Fig. 12 and Supplementary Data 1–3). Labeling the cells with Pr$_2$GalNAz at 100 μM, one-half of the Ac$_2$GalNAz concentration, already resulted in identification of 699 O-GlcNAc sites and 10 S-glycosylation sites (Fig. 2g, Supplementary Fig. 13 and Supplementary Data 4–6). As previously reported[8], 312 O-GlcNAc sites and 19 S-GlcNAc sites were identified in HeLa cells treated with 1 mM GalNAz. Of the 312 O-GlcNAc sites, 87% were covered by the Pr$_2$GalNAz-identified list. Furthermore, Cys 1139 of host cell factor 1 (HCF1), a previously identified S-GlcNAc site in HEK cells[21], was also identified by Pr$_2$GalNAz. These results establish that Pr$_2$GalNAz possesses extremely high efficiency for metabolic labeling of O-GlcNAc and does not induce artificial S-glycosylation.

**ESRRB O-GlcNAcylation in mESCs revealed by Pr$_2$GalNAz**. We next applied Pr$_2$GalNAz to investigate O-GlcNAcylation in mouse embryonic stem cells (mESCs). Derived from the inner cell mass of blastocysts, ESCs are pluripotent cells that can self-renew indefinitely and differentiate into all cell types of the body, thus holding great potential for regenerative medicine[22,23]. Emerging evidence suggests that O-GlcNAc is important for ESC maintenance. *Ogt* gene knockout in mice is embryonically lethal, and OGT is essential for stem cell viability and somatic cell survival[24,25]. Global blocking of O-GlcNAcylation in mESCs hampers self-renewal, and increasing the O-GlcNAc level inhibited differentiation[26–30]. Maintenance of mESC pluripotency is regulated by a transcription factor (TF) network centered on three master TFs, OCT4, SOX2, and NANOG (OSN)[31,32]. OCT4 and SOX2 are both O-GlcNAcylated in mESCs[27,33,34]. OCT4 O-GlcNAcylation regulates its transcriptional activity, hence facilitating pluripotency maintenance[27]. Conversely, O-GlcNAcylation on SOX2 was shown to inhibit pluripotency[34]. The auxiliary TFs of the network, such as ESRRB, KLF2, KLF4, and TBX3, are also important for pluripotency[35]. Considering the broad protein substrates of OGT found in HeLa cells, we wondered whether the auxiliary TFs in mESCs could also be O-GlcNAc modified.

To test this hypothesis, mESCs were incubated with Pr$_2$Gal-NAz for 48 h, reacted with alkyne-PEG$_4$-biotin, and captured with streptavidin beads. Immunoblot analysis exhibited significant enrichment of ESRRB, indicating that ESRRB was modified by O-GlcNAz (Fig. 3a). To further validate the Pr$_2$GalNAz-labeling results, the mESC lysates were treated with a mutant galactosyltransferase (Y298L GalT1), which recognizes terminal GlcNAc, including O-GlcNAc, and attaches a GalNAz moiety using UDP-GalNAz[36]. Click-labeling with alkyne-PEG$_4$-biotin,

followed by streptavidin bead pulldown, confirmed O-GlcNAc modification on endogenous ESRRB (Fig. 3b). The O-GlcNAcylation stoichiometry of ESRRB was quantified by labeling O-GlcNAc with a resolvable mass tag[37], alkyne-functionalized polyethylene glyco 2000 (Alkyne-PEG$_{2kD}$), which showed a modification ratio of approximately 52% (Fig. 3c). Furthermore, by using a FLIM-FRET-based imaging method[38], the O-GlcNAcylation on ESRRB was visualized in situ in HeLa cells (Supplementary Fig. 14). In addition, lysates of HEK293T cells overexpressing FLAG-ESRRB and EGFP-OGT were co-immunoprecipitated with an anti-FLAG antibody. EGFP-OGT was detected in the co-immunoprecipitates, indicating direct binding between OGT and ESRRB (Fig. 3d). Taken together, these results demonstrate that ESRRB is a bona fide O-GlcNAcylated protein in mESCs.

To identify the O-GlcNAc modification sites of ESRRB, His-tagged ESRRB was expressed and purified from *Escherichia coli* (*E. coli*) co-expressing OGT[39]. LC-MS/MS with ETD fragmentation analysis on the purified ESRRB identified the peptide with amino acid 16-30 (IKTEPSSPSSGIDAL) as an O-GlcNAcylated peptide. The modification site was mapped to Ser 24 or Ser 25 (Fig. 3e). Notably, ESRRB Ser 24 and Ser 25 in mouse are well conserved in rat and human (Supplementary Fig. 15). To further determine the exact site, the ESRRB mutants, ESRRB$^{S24A}$, ESRRB$^{S25A}$, and ESRRB$^{S24AS25A}$ were expressed and purified. Immunoblotting with an O-GlcNAc-specific antibody RL2 detected O-GlcNAcylation of wild-type ESRRB, and interestingly a stronger O-GlcNAc signal on ESRRB$^{S24A}$ (Fig. 3f). Moreover, the RL2 binding was competed off by 1 M GlcNAc. Both ESRRB$^{S25A}$ and ESRRB$^{S24AS25A}$ showed minimal RL2 binding, suggesting that Ser 25 is a major O-GlcNAc modification site for ESRRB. Significant loss of O-GlcNAcylation for ESRRB$^{S25A}$ was further validated in HEK293T cells by two independent O-GlcNAc detection methods, Pr$_2$GalNAz-labeling and Y298L GalT1-based chemoenzymatic labeling (Fig. 3g, h). To confirm the O-GlcNAcylation site of ESRRB in mESCs, mESCs stably expressing FLAG-ESRRB and FLAG-ESRRB$^{S25A}$ were generated, in which the exogenous ESRRB mRNA and protein expression levels were ~1–2 folds of the endogenous levels (Supplementary Fig. 16). As assayed by chemoenzymatic labeling, FLAG-ESRRB$^{S25A}$ exhibited a significantly reduced labeling intensity than FLAG-ESRRB, indicating that ESRRB Ser 25 was O-GlcNAcylated in mESCs (Fig. 3i).

**O-GlcNAc regulates ESRRB stability and activity**. Given that O-GlcNAc was found to stabilize a number of proteins[13,40,41], we first tested whether the expression level of ESRRB is regulated by O-GlcNAcylation. As shown by quantitative RT-PCR (qRT-PCR), OGT inhibition by Ac$_4$5SGlcNAc did not alter the mRNA level of ESRRB (Supplementary Fig. 17). The protein synthesis was blocked by cycloheximide (CHX) in mESCs stably expressing FLAG-ESRRB, and immunoblot analysis showed that protein degradation of ESRRB was significantly accelerated upon OGT inhibition (Fig. 4a). Furthermore, FLAG-ESRRB$^{S25A}$ stably expressed in mESCs degraded faster than FLAG-ESRRB (Fig. 4b). Similar results were observed by transiently expressing EGFP-ESRRB and EGFP-ESRRB$^{S25A}$ with an equal amount of plasmid in HEK293T cells (Fig. 4c). In addition, the accelerated degradation of ESRRB$^{S25A}$ was correlated with increased ubiquitination (Supplementary Fig. 18). These results demonstrate that ESRRB O-GlcNAcylation enhances its protein stability.

In mESCs, ESRRB can directly bind to OCT4 and NANOG, which plays important roles in regulating their transcription activity and sustaining self-renewal[42,43]. We therefore asked whether ESRRB O-GlcNAcylation could affect its interactions

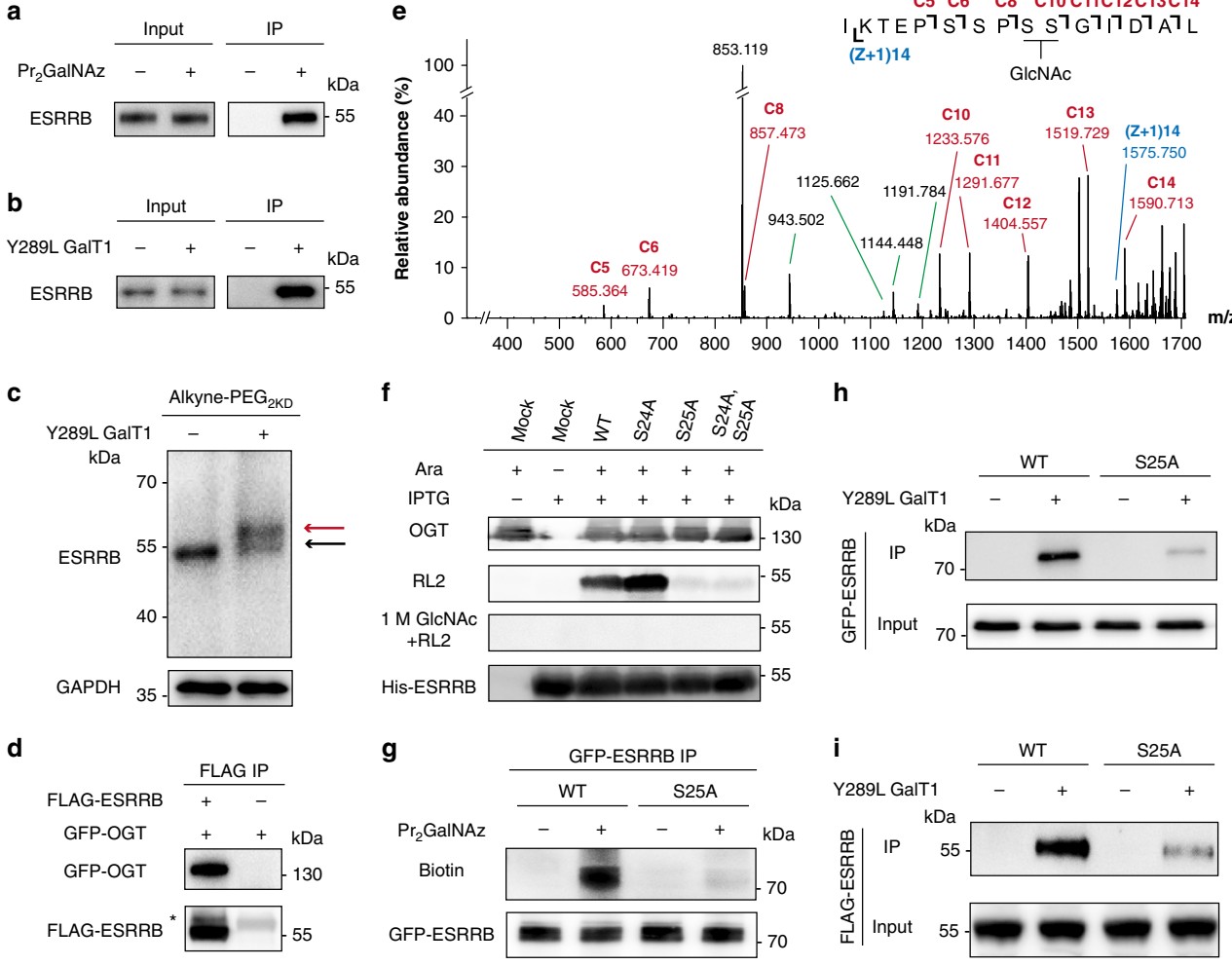

**Fig. 3** ESRRB is O-GlcNAcylated at Ser 25. **a** Immunoblots showing mESC R1 cell lysates incubated with Pr₂GalNAz, lysed, reacted with alkyne-PEG₄-biotin, and captured by streptavidin beads. **b** Immunoblots showing mESC R1 cell lysates incubated with Y289L GalT1 and UDP-GalNAz, reacted with alkyne-PEG₄-biotin, and captured by streptavidin beads. **c** Immunoblots showing mESC R1 cell lysates incubated with Y289L GalT1 and UDP-GalNAz, and reacted with Alkyne-PEG₂ₖD. Anti-GAPDH blot demonstrates comparable loading. **d** Immunoblots showing HEK293T cells expressing GFP-OGT or co-expressing GFP-OGT and FLAG-ESRRB co-immunoprecipitated with an anti-FLAG antibody. The asterisk indicated IgG heavy chain. **e** ETD MS/MS spectrum of the ESRRB peptide with an O-GlcNAcylation site. The matched fragment ions are labeled in red and blue. C8 and C10 ions indicate that the O-GlcNAc is located on Ser 24 or Ser 25. **f** Immunoblots showing *E. coli* co-expressing OGT (under control of an arabinose-inducible promoter) and His-ESRRB (IPTG-inducible) with single or double mutations. Ara, arabinose. RL2 is an O-GlcNAc-specific antibody. Pre-treatment with 1 M GlcNAc was used to block RL2. **g** Streptavidin blots showing HEK293T cells expressing EGFP-ESRRB or EGFP-ESRRB[S25A] incubated with Pr₂GalNAz for 48 h, lysed, and immunoprecipitated with an anti-GFP antibody. Anti-GFP blot demonstrates comparable loading. **h**, **i** Immunoblots showing HEK293T cell lysates overexpressing EGFP-ESRRB or EGFP-ESRRB[S25A] (**h**) and mESC cell lysates stably expressing FLAG-ESRRB or FLAG-ESRRB[S25A] (**i**) incubated with Y289L GalT1 and UDP-GalNAz, reacted with alkyne-PEG₄-biotin, and captured by streptavidin beads. In **a–i** representative results are shown from three independent experiments. Source data for figures **a–i** are provided as a Source Data file

with OCT4 and NANOG. mESCs stably expressing FLAG-ESRRB and FLAG-ESRRB[S25A] were lysed, immunoprecipitated with an anti-FLAG antibody, and the co-immunoprecipitated endogenous OCT4 and NANOG were detected by immunoblotting, which showed significantly less binding of OCT4 and NANOG to ESRRB[S25A] than wild-type ESRRB (Fig. 4d). Similar results were detected by co-expressing EGFP-ESRRB/EGFP-ESRRB[S25A] with FLAG-OCT4 or FLAG-NANOG in HEK293T cells (Fig. 4e, f). To further validate O-GlcNAc regulation on those protein-protein interactions, the O-GlcNAc level in HEK293T cells was raised by an OGA inhibitor, Thiamet-G[44]. Co-immunoblot analysis showed obviously enhanced binding of OCT4 and NANOG to ESRRB upon OGA inhibition (Fig. 4g, h).

O-GlcNAcylation occurs in the AF-1 domain of ESRRB, which confers basic transactivation[45,46]. We therefore investigated

whether ESRRB O-GlcNAcylation could alter ESRRB transcriptional activity. An equal amount of ESRRB-responsive element-driven luciferase and FLAG-ESRRB plasmids were co-transfected in HEK293T cells. Upon OGT inhibition with Ac₄5SGlcNAc, the transcriptional activity of ESRRB was significantly suppressed as shown by the luciferase assay (Fig. 4i). Furthermore, the S25A mutant of ESRRB exhibited decreased transcriptional activity (Fig. 4j). In addition, RNA sequencing analysis on mESCs stably expressing FLAG-ESRRB[WT] and FLAG-ESRRB[S25A] revealed 37 differentially expressed genes, of which 25 were previously identified as ESRRB target genes[47] (Supplementary Fig. 19 and Supplementary Data 7). Together, these results demonstrate that Ser 25 O-GlcNAcylation on ESRRB regulates its protein stability, interactions with OCT4 and NANOG, and its transcriptional activity.

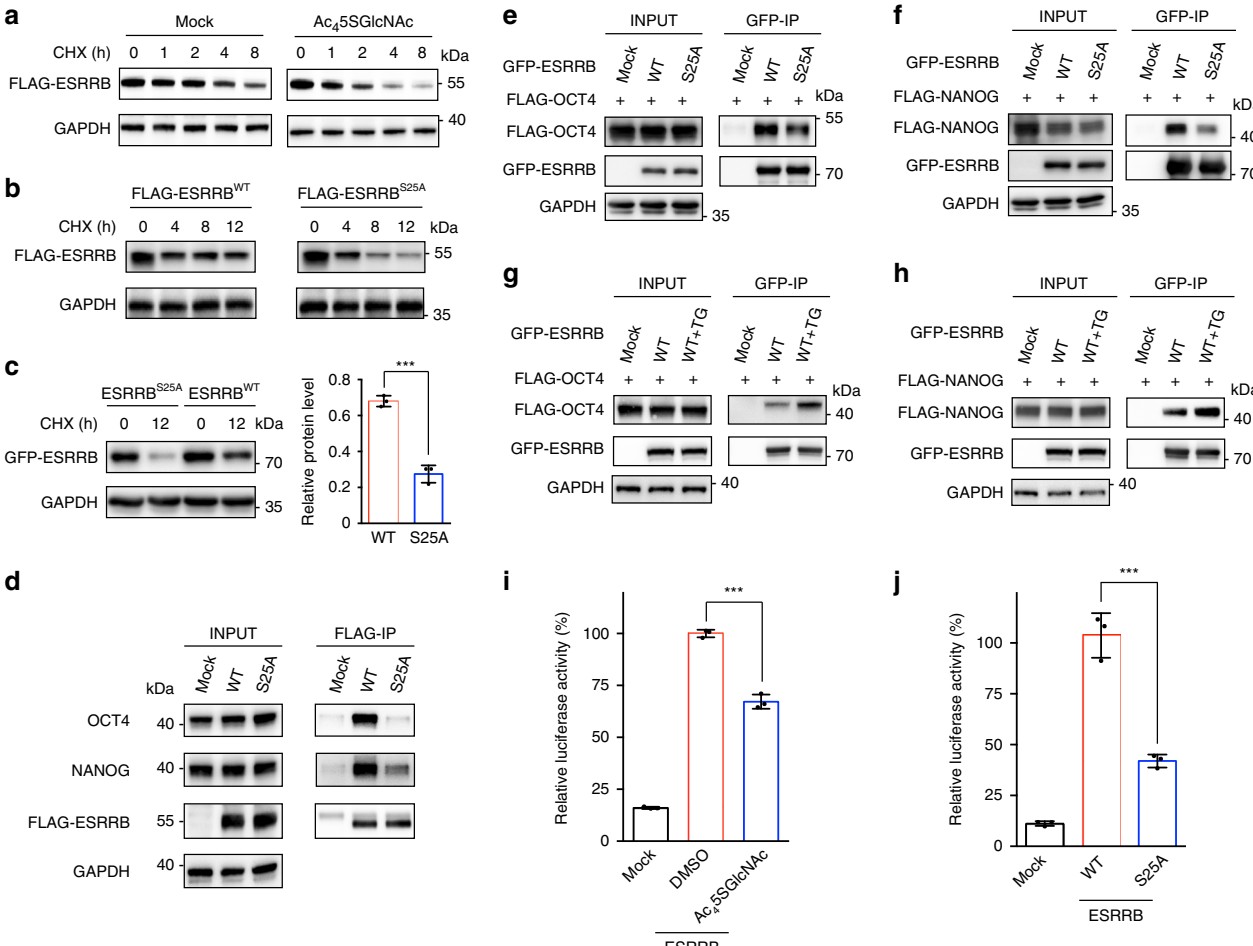

**Fig. 4** ESRRB O-GlcNAcylation enhances protein stability, protein-protein interactions and transcriptional activity. **a** Immunoblots showing expedited degradation of ESRRB upon O-GlcNAcylation inhibition. mESCs expressing FLAG-ESRRB pre-treated with or without Ac$_4$5SGlcNAc were incubated with 50 μM CHX for up to 8 h, during which the expression of FLAG-ESRRB was monitored. **b** Immunoblots showing mESCs expressing FLAG-ESRRB or FLAG-ESRRB[S25A] treated with CHX for up to 12 h. **c** Immunoblots showing HEK293T cells expressing EGFP-ESRRB or EGFP-ESRRB[S25A] treated with CHX for 12 h. Bar graph showing statistical analysis of the relative protein level of HEK293T cells treated with CHX for 12 h (normalized to that with no CHX). Error bars represent mean ± s.d. ***$P < 0.001$ (Student's $t$-test). **d** Immunoblots showing co-immunoprecipitation of endogenous OCT4 and NANOG with ESRRB in mESCs. mESCs expressing FLAG-ESRRB or FLAG-ESRRB[S25A] were immunoprecipitated with an anti-FLAG antibody. **e**, **f** Immunoblots showing co-immunoprecipitation of OCT4 and NANOG with ESRRB in HEK293T cells. The cells co-expressing FLAG-OCT4 with EGFP-ESRRB or EGFP-ESRRB[S25A] or co-expressing FLAG-NANOG with EGFP-ESRRB or EGFP-ESRRB[S25A] were immunoprecipitated with an anti-GFP antibody. **g**, **h** Increasing O-GlcNAcylation promoted interactions of OCT4 and NANOG with ESRRB. With the treatment of DMSO or Thiamet-G (TG) for 48 h, HEK293T cells co-expressing FLAG-OCT4 with EGFP-ESRRB or co-expressing FLAG-NANOG with EGFP-ESRRB were immunoprecipitated with an anti-GFP antibody. **i** Luciferase assay showing reduced ESRRB transcriptional activity upon O-GlcNAc inhibition. **j** Luciferase assay showing transcriptional activities of WT and S25A mutant ESRRB. In **i** and **j** results are shown from three independent experiments. ***$P < 0.001$ (one-way ANOVA). The anti-GAPDH gels in **a**–**f** demonstrate comparable loading. In **a**–**h** representative results are shown from three independent experiments. Source data for figures **a**–**j** are provided as a Source Data file

**ESRRB O-GlcNAcylation enhances mESC pluripotency.** Within the TF network, ESRRB is important for maintaining the pluripotency of mESCs. ESRRB is a target of NANOG and the GSK3 signaling pathway. Under the culture conditions with serum but without leukemia inhibitory factor (LIF), constitutive overexpression of ESRRB can revert epiblast-derived stem cells (EpiSCs) to an ESC-like state and maintain mESC self-renewal and pluripotency[47,48]. To investigate whether O-GlcNAc regulates ESRRB functions in this context, mESCs stably expressing FLAG-ESRRB and FLAG-ESRRB[S25A] were maintained in serum without LIF for 6 days and 9 days, and the undifferentiated state was detected by alkaline phosphatase (AP) staining (Fig. 5a and Supplementary Fig. 20a, b). In agreement with previous results[47], overexpression of wild-type (WT) ESRRB promoted formation of AP-positive colonies, which included undifferentiated and mixed

colonies. In contrast, ESRRB[S25A] resulted in significantly less undifferentiated and mixed colonies. Furthermore, the proportion of AP-positive colonies in ESRRB[S25A]-expressing mESCs decreased more rapidly up to 9 d than mESCs expressing WT ESRRB (Supplementary Fig. 20c). Accordingly, the expression level of mESC marker genes including *Oct4*, *Nanog*, and *Rex1* decreased more rapidly in cells expressing ESRRB[S25A] than the WT ESRRB-expressing cells upon removing LIF from the culture medium (Supplementary Fig. 21).

When implanted into immunodeficient hosts, ESCs with self-renewal capability and pluripotency can differentiate to form teratomas, a tumor-like formation consisting of all three germ layers[22,49]. We therefore tested whether ESRRB O-GlcNAcylation affects the teratoma-forming potential of mESCs in immunodeficient mice. Implantation of mESCs stably expressing

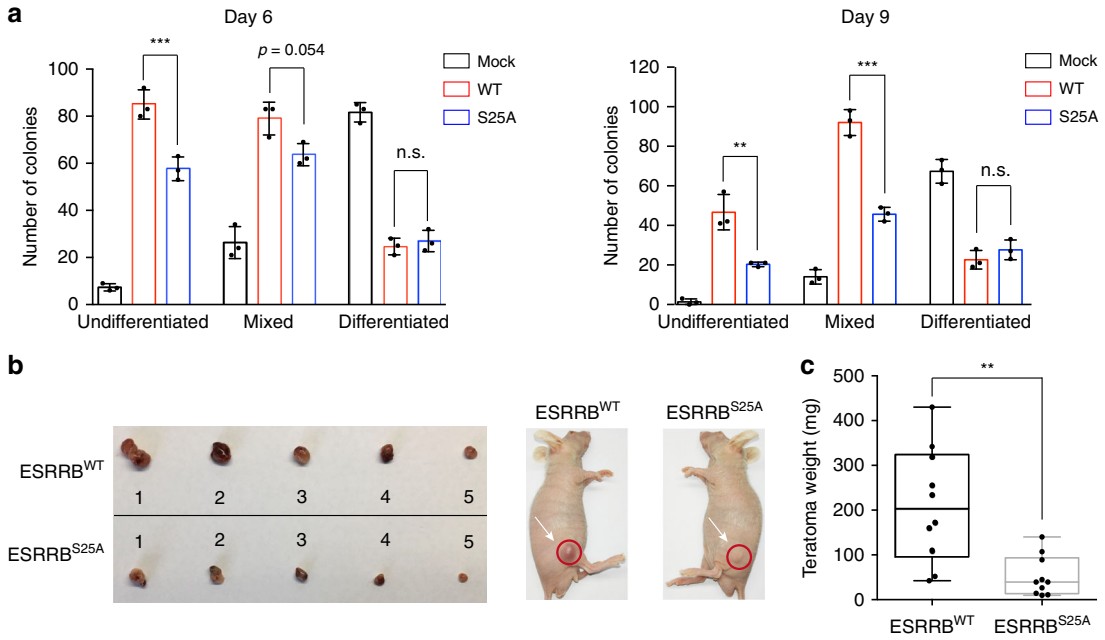

**Fig. 5** ESRRB O-GlcNAcylation enhances self-renewal and pluripotency. **a** Clonogenicity assay of mESCs stably expressing Mock, FLAG-ESRRB[WT] and FLAG-ESRRB[S25A]. Bar graphs showing the statistical analysis of the undifferentiated, mixed and differentiated colonies in the absence of LIF for 6 days (left panel) and 9 days (right panel). Error bars represent mean ± s.d. Results are from three independent experiments. **P < 0.01, ***P < 0.001, n.s., not significant (one-way ANOVA). **b** Teratoma formation in immunodeficient mice by mESCs expressing FLAG-ESRRB or FLAG-ESRRB[S25A]. **c** Box plot showing the statistical analysis of teratoma weight. Median and quartile values are provided by the central line and box boundaries. Whiskers show min to max values. Results are from ten independent experiments. **P < 0.01 (Student's t-test). Source data for figures **a**, **c** are provided as a Source Data file

FLAG-ESRRB[S25A] gave rise to teratomas with significantly reduced sizes, comparing to those formed by mESCs stably expressing FLAG-ESRRB (Fig. 5b, c and Supplementary Fig. 22). Furthermore, the tumors generated by mESCs expressing WT or S25A ESRRB exhibited similar germ layers including the ectoderm, mesoderm, and endoderm (Supplementary Fig. 23). These results suggest that ESRRB O-GlcNAcylation regulates proliferation and self-renewal of mESCs during teratoma formation.

## Discussion

Given the broad usage of clickable unnatural monosaccharides for imaging and profiling of cellular glycosylation, the existence of the per-O-acetylation-induced non-specific S-glycosylation has raised the concern that some false-positives might have been overlooked in previous studies using per-O-acetylated sugars. Although the extent of artificial S-glycosylation and how much interference it may cause probably vary case by case, improving the unnatural sugar reporters to avoid artificial S-glycosylation is desired and highly valuable for the field. In this study, we developed the next-generation unnatural monosaccharides for MGL based on the hypothesis that partial O-esterification might avoid S-glycosylation while preserving enough membrane-permeability.

We demonstrated that 1,3-di-O-esterification is a satisfying protection strategy for developing next-generation unnatural sugars with no artificial S-glycosylation for MGL. Two partially protected GalNAz, 1,3-Ac$_2$GalNAz, and 1,3-Pr$_2$GalNAz, exhibited significantly improved labeling efficiency compared to the unprotected GalNAz. In particular, 1,3-Pr$_2$GalNAz, by using the propionyl group, a protecting group with a longer alkyl chain, to compensate for the loss of the number of protecting groups, possesses a superb metabolic labeling efficiency. Another benefit of the propionyl group is its low tendency for acyl group

migration from the 3-hydroxyl group to the 4- and 6-hydroxyl groups. Owing to the migration of 3-O-acetyl group, 1,3-Ac$_2$GalNAz may spontaneously transform to a mixture containing 1,3-Ac$_2$GalNAz, 1,4-Ac$_2$GalNAz, and 1,6-Ac$_2$GalNAz in the culture medium. This transformation probably should not significantly affect the labeling specificity and efficiency. Nevertheless, 1,3-Pr$_2$GalNAz is a better optimized next-generation unnatural sugar for MGL.

It should be noted that the tri-O-acetylated GalNAz, 3,4,6-Ac$_3$GalNAz, has higher reactivity toward Cys, suggesting that deacetylation at the anomeric oxygen of Ac$_4$GalNAz is required for non-enzymatic S-glycosylation and might be a rate-limiting step. Furthermore, the fact that 1,3,6-Ac$_3$GalNAz has slight S-glycosylation demonstrates that di-O-esterification is critical for completely eliminating the side reaction.

3,4,6-Ac$_3$GalNAz induced severe cytotoxicity, and Ac$_4$GalNAz at high concentrations was moderately toxic. These results raised the possibility that S-glycosylation is a causative factor for cell death. In support of this hypothesis, GalNAz, Ac$_2$GalNAz, and Pr$_2$GalNAz with no S-glycosylation were not cytotoxic. Interestingly, various esterified hexosamine molecules have been previously evaluated for anti-cancer activity[50–54]. In agreement with our results, 3,4,6-tri-O-butanoylated ManNAz showed the highest inhibition of cancer cell metastasis[50]. Our results warrant future investigation on critical Cys residues that can be targeted by S-glycosylation in the context of cell death pathways and anti-cancer drug discovery.

As a showcase of MGL using Pr$_2$GalNAz, we identified ESRRB as an O-GlcNAc modified protein in mESCs. The pluripotency TFs form an interconnected network, which involves various protein-protein interactions and cooperative binding to adjacent DNA sites[55]. OGT has been found to modify multiple components of the same protein complexes[13]. It is appealing to hypothesize that O-GlcNAc regulates mESC pluripotency by targeting

TFs at the network level. In addition to several previously identified O-GlcNAcylated TFs in mESCs, including OCT4 and SOX2[27,34], our work adds ESRRB to the list, supporting this hypothesis.

While O-GlcNAcylation enhances OCT4 activity and inhibits SOX2 activity[27,34], we found that O-GlcNAcylation regulates the function of ESRRB by mediating the protein stability. It appears that ESRRB O-GlcNAcylation at Ser 25 inhibits ubiquitination, thus enhancing protein stability. Moreover, O-GlcNAcylation also significantly enhances binding of ESRRB to OCT4 and NANOG. Of note, it was previously reported that ESRRB binds NANOG through its DNA binding domain[42]. The Ser 25 O-GlcNAcylation resides in the AF-1 domain. ESRRB O-GlcNAc may affect interactions between ESRRB and NANOG through regulating protein stability or/and conformation. By stabilizing ESRRB, promoting its interactions with master pluripotency TFs, and facilitating its transcriptional activity, O-GlcNAc enhances the ESRRB activity in maintaining mESC pluripotency and self-renewal. Given previous examples on O-GlcNAcylation stabilizing modified proteins and protein complex formation[13], it will be interesting to test whether enhancement of protein stability by O-GlcNAcylation is a generic mechanism for the pluripotency TF network in mESCs.

O-GlcNAc has been reported to interplay with phosphorylation[56,57]. Phosphorylation sites of ESRRA and ESRRG, two members in the estrogen related receptors family, were previously identified[58–60]. For example, phosphorylation at Ser 19 of ESRRA regulated its transcriptional activity and its SUMOylation at Lys 14[58,59]. Although the corresponding Ser and Lys residues in ESRRB are near Ser 25, no phosphorylation site in close proximity to the O-GlcNAc site has been experimentally identified[61]. It will be interesting to investigate whether ESRRB O-GlcNAcylation has crosstalk with other posttranslational modifications.

O-GlcNAc is emerging as a key regulator for ESC pluripotency and self-renewal. Transcriptional control is the most critical mechanism for regulation of the ESC state[55]. One of the best studied functions of O-GlcNAc is to regulate transcription through modification of TFs, which has been investigated in various cancer cells[9]. Investigation of O-GlcNAc functions in stem cells is in its infancy. O-GlcNAcylation on TFs seems to be a rational choice for detailed studies. The next-generation unnatural monosaccharides developed in this work should find broad applications in stem cells as well as in other cell types and organisms. Finally, the 1,3-diesterification strategy can be readily applied for developing next-generation chemical reporters based on other unnatural sugars, such as ManNAz for probing other types of glycosylation.

## Methods

**Compound synthesis**. The synthesis of compound **2**, **3**, and **4** is described in Supplementary Note 1 in the Supplementary Information.

**Cell culture**. R1 murine embryonic stem cells (mESCs) were grown on 0.1% gelatin (Sigma-Aldrich) coated cell culture dishes (Invitrogen). Cells were cultured in the KnockOut™ DMEM (Gibco) supplemented with 15% FBS (Hyclone), 1 × non-essential amino acids (Gibco), 1 × GlutaMAX™ (Gibco), 1000 U/mL leukemia inhibitory factor (LIF) (Millipore), 100 μM 2-mercaptoethanol (Sigma-Aldrich), 100 U/mL penicillin, and 100 μg/mL streptomycin (Gibco). mESCs were disassociated with 0.1% trypsin (Gibco) and passaged at a split ratio of 1 to 15 every three days. R1 mESCs were kindly provided by Prof. Ye-Guang Chen at Tsinghua University. HEK 293FT cells were kindly provided by Prof. Qin Shen at Tongji University. HEK 293FT, HeLa (ATCC® CCL-2), CHO (ATCC® CCL-61), Neuro-2a (ATCC® CCL-131), HT-1080 (ATCC® CCL-121), SH-SY5Y (ATCC® CRL-2266), NCI-H1299 (ATCC® CRL-5803), A549 (ATCC® CCL-185), NIH/3T3 (ATCC® CRL-1658), MCF-7 (ATCC® HTB-22), HEK293T (ATCC® CRL-11268) cells were obtained from American Type Culture Collection and cultured in DMEM (Gibco) supplemented with 10% FBS (Gibco), 100 U/mL penicillin, and 100 μg/mL streptomycin (Gibco). Cells in all experiments were within 20 passages and free of mycoplasma contamination.

**Plasmids and transfection**. Full-length *Nanog* was subcloned from R1 mESC cDNA into p3xFLAG-CMV-10 with the forward primer 5′-CCCAAGCTTATG AGTGTGGGTCTTCCTGG-3′ and the reverse primer 5′-GGGGTACCTCATA TTTCACCTGGTGGAGTCAC-3′; Full-length *Oct4* was subcloned from R1 mESC cDNA into p3xFLAG-CMV-10 with the forward primer 5′-GGAATTCAATG GCTGGACACCTGGC-3′ and the reverse primer 5′-GGGGTACCTCAGTTTG AATGCATGGGAGAGC-3′; Full-length *Esrrb* was subcloned from R1 mESC cDNA into p3xFLAG-CMV-10 or pEGFP-C1 with the forward primer 5′-CGG AATTCAATGCTGCTGAACCGAATGTC-3′ and the reverse primer 5′-CGG GATCCTCACACCTTGGCCTCCA-3′; Full-length *Ogt* was inserted into pEGFP-N1 with the forward primer 5′-ACGCGTCGACGGATGGCGTCTTCCGTGGG-3′ and the reverse primer 5′-CGGGATCCTTATGCTGACTCAGTGACTTCAAC AG-3′; Full-length *Esrrb* was subcloned from R1 mESC cDNA into pET-28a(+) with the forward primer 5′-CGGGATCCATGCTGCTGAACCGAATGTC-3′ and the reverse primer 5′-CGGAATTCTCACACCTTGGCCTCCA-3′; Full-length *Esrrb*^S24A was inserted to pET-28a(+) with the forward primer 5′-GGAGCCAT CCAGCCCGGCCTCGGGCAT-3′ and the reverse primer 5′-CCGGGCTGGA TGGCTCCGTCTTGATGA-3′; Full-length *Esrrb*^S25A was inserted to pET-28a(+) or pEGFP-C1 with the forward primer 5′-GCCATCCAGCCCGTCCGCGGGCA TTGA-3′ and the reverse primer 5′-CGGACGGGCTGGATGGCTCCGTCTTGA-3′; Full-length *Esrrb*^S24AS25A was inserted to pET-28a(+) with the forward primer 5′-GCCATCCAGCCCGGCCGCGGGGCATTGA-3′ and the reverse primer 5′-CG GCCGGGCTGGATGGCTCCGTCTTGA-3′; Full-length *Flag-Esrrb* was subcloned from R1 mESC cDNA into pCDH-EF1-MCS-T2A-copGFP with the forward primer 5′-GGAATTCATGGACTACAAAGACGATGACGACAAGATGTCGTC GAAGACAGGC-3′ and the reverse primer 5′-ACGCGTCGACCACCTTGGCCTC CAGCATC-3′; Full-length *Flag-Esrrb*^S25A was subcloned from pEGFP-ESRRB^S25A to pCDH-EF1-MCS-T2A-copGFP with the forward primer 5′-GCCATCCAGCC CGTCCGCGGGCATTGA-3′ and the reverse primer 5′-CGGACGGGCTGGA TGGCTCCGTCTTGA-3′. Transfections of all plasmids in this study were performed using VigoFect (Vigorous Biotechnology) based the manufacturer's instructions. ESRRB-responsive element-driven luciferase reporter plasmid (11569ES03) was purchased from Yeasen Biotechnology.

**Antibodies, reagents, and equipment**. Antibodies included anti-ESRRB (Abclonal, A13977, 1:500), anti-DDDDK-tag (MBL, M185-3, 1:5,000), anti-GFP (Abcam, ab183734, 1:5,000), anti-HIS (CST, 12698, 1:3,000), anti-RL2 (Abcam, ab2739, 1:1,000), anti-OGT (Abcam, ab177941, 1:2,000), anti-HA (CST, 3724, 1:3,000), anti-GAPDH-HRP (Sigma-Aldrich, G9295, 1:10,000), anti-streptavidin-HRP (Beyotime, A0303, 1:3,000), anti-mouse IgG-HRP (Abcam, ab97023, 1:5000), anti-rabbit IgG-HRP (Abcam, ab6721, 1:5,000). Antibody-conjugated magnetic beads for immunoprecipitation were anti-DDDDK-tag (FLAG tag) mAb-Magnetic Beads (MBL, M185-11) and anti-GFP-mAb-Magnetic Beads (MBL, D153-11). Alkyne-PEG₂ₖᴅ (catalog no. 699802), MG-132 (catalog no. M7449) and CHX (catalog no. 01810) were obtained from Sigma. Alkyne-PEG₄-biotin (catalog no. TA105), alkyne-Cy5 (catalog no. TA116), alkyne-Cy3 (catalog no. TA117) and alkyne-TAMRA (catalog no. TA108) were purchased from Click Chemistry Tools. Monosaccharides were purchased from Carbosynth, trifluoroacetic acid was purchased from Acros Organics and other compounds used in the synthesis were from J&K Scientific. Compound **1**, 2-[4-{(bis[(1-tert-butyl-1H-1,2,3-triazol-4-yl)methyl] amino)methyl}-1H-1,2,3-triazol-1-yl]acetic acid (BTTAA) and acid-cleavable biotin tags (alkyne-AC-biotin) were synthesized[8,20,62]. In the LC-MS/MS analysis, dithiothreitol (DTT) was purchased from J&K Scientific, iodoacetamide was purchased from Sigma-Aldrich, and sequencing-grade modified trypsin and trypsin resuspension buffer were purchased from Promega. All of the organic solvent was used as analytic grade or better. LC-MS detection was performed using Waters ACQUITY UPLC I-Class SQD 2 MS spectrometer with electrospray ionization (ESI). ¹H NMR, ¹³C NMR, COSY, HSQC and HMBC spectra were recorded on a Bruker-500 MHz NMR (AVANCE III) instrument. Compound **9** was further purified by HPLC (XBridge Prep C18, 5 μm, OBD 30 × 25 mm column). High-resolution mass spectra (HRMS) were recorded on a Fourier Transform Ion Cyclotron Resonance Mass Spectrometer (APEX IV). The imaging of SDS-PAGE gels and western blotting membranes were acquired from ChemiDoc XRS+(Bio-Rad) and Tanon-5200Multi (Tanon).

**Cell viability assay**. About 2000 cells per well plate containing 100 μL medium were plated on a 96-well, cultured overnight, and treated with vehicle or the indicated unnatural sugar at varied concentrations for 48 h. Ten microliters 2-(2-methoxy-4-nitrophenyl)-3-(4-nitrophenyl)-5-(2,4-disulfophenyl)-2H-tetrazolium, monosodium salt (WST-8, Enhanced Cell Counting Kit-8, Beyotime, C0042) was then added to each well, and the cells were further incubated at 37 °C for 1 h. The absorbance of 450 nm was measured and normalized to the vehicle-treated cells.

**In vitro assays for the artificial cysteine reaction**. For reaction with cell lysates, 50 μL lysates (2 mg/mL in PBS, pH 7.4) obtained by sonication were incubated with indicated unnatural sugars at varied concentrations at 37 °C for 2 h, followed by

precipitation by adding 150 μL methanol, 37.5 μL chloroform and 100 μL Milli-Q water. After centrifugation (18,000 g, 5 min) and removal of the aqueous phase, 100 μL methanol was added. For reaction with purified proteins, 50 μL of 1 mg/mL CFL1, GAPDH or PRDX1 in PBS was incubated with 1 mM indicated unnatural sugars at 37 °C for 2 h, followed by addition of 150 μL methanol, 37.5 μL chloroform and 100 μL Milli-Q water, centrifugation, removing the aqueous phase, and adding 100 μL methanol. The methanol was then removed by centrifugation and the resulting lysates or proteins were re-suspended in 50 μL 0.4% SDS-PBS (wt/vol) for In-gel fluorescence scanning. For pre-blocking Cys-glycosylation in HeLa cell lysates (2 mg/mL in PBS, pH 7.4), iodoacetamide was added (25 mM as final concentration), incubated at 37 °C for 1 h, subjected to in vitro reaction with cell lysates, and used for In-gel fluorescence scanning. For analysis of purified CFL1 reacting with different unnatural sugar probes, the reaction system was detected by LC-MS on an ACQUITY UPLC I-Class SQD 2 MS spectrometer with a BEH300 C4 Acquity column (1.7 μm, 2.1 × 100 mm) using a 16 min gradient (5%-90% acetonitrile, 0.4 mL/min). For analysis of glutathione reactions, 50 μL glutathione (5 mM in PBS, pH 7.4) was incubated with indicated unnatural sugars at varied concentrations at 37 °C for 48 h. The system was then diluted with 500 μL methanol, followed by analyzing by LC-MS on an ACQUITY UPLC I-Class SQD 2 MS spectrometer with a BEH C18 Acquity column (1.7 μm, 2.1 × 50 mm) using a 6 min gradient (5–95% acetonitrile, 0.3 mL/min).

**Metabolic labeling of live cells**. The cells at 30% confluence in sixwell plates or 15-cm dishes were treated with indicated unnatural sugars at varied concentrations for 48 h. The cells were harvested by trypsin digestion and washed for three times with PBS. The cell pellets were re-suspended with cold RIPA buffer [1% Nonidet P-40 (vol/vol), 1% sodium deoxycholate (wt/vol), 150 mM NaCl, 0.1% SDS (wt/vol), 50 mM triethanolamine, EDTA-free protease inhibitor mixture (Pierce), pH 7.4] and lysed with sonication, followed by centrifuging at 12,000 g for 10 min at 4 °C to remove debris and adjusting the protein concentration by BCA protein assay kit (Pierce) to 2 mg/mL. The suspensions of lysates from the 6-well plates and 15-cm dishes were used for In-gel fluorescence scanning or Enrichment of O-GlcNAcylated proteins and on-bead digestion, respectively. For OGT inhibitor treatment experiment in HeLa cells, the cells at 30% confluence in six-well plates were co-treated with 50 μM Ac4SGlcNAc and indicated unnatural sugars at varied concentrations for 48 h, harvested as mentioned above, and used for In-gel fluorescence scanning.

**Chemoenzymatic labeling of O-GlcNAcylated proteins**. The cells in six-well plates or 15-cm dishes were lysed with RIPA buffer and the lysates were re-suspended in the buffer containing 20 mM Hepes buffer (pH 7.9) containing 1% SDS (wt/vol) with a final concentration of 1 mg/mL. To 100 μL lysates, were added orderly 24.5 μL Milli-Q water, 40 μL labeling buffer [125 mM NaCl, 5% Nonidet P-40 (vol/vol), 50 mM Hepes, pH 7.9], 5.5 μL 100 mM MnCl2, 5 μL 500 μM UDP-GalNAz, and 3.75 μL Y289L GalT1. In the negative control, Y289L GalT1 was omitted. The mixture was softly rotated at 4 °C for 20 h and the proteins were precipitated as described above. The proteins were re-suspended in 50 μL 50 mM Tris-HCl (pH 8.0) containing 1% SDS (wt/vol) for In-gel fluorescence scanning. For enrichment of O-GlcNAc proteins, the procedures were scaled up with the starting material of 2 mL 1 mg/mL lysates, and the resulting 1 mL of 2 mg/mL protein suspensions were subjected to Enrichment of O-GlcNAcylated proteins and on-bead digestion.

**In-gel fluorescence scanning**. For visualizing the in vitro reacted samples, 50 μL suspensions obtained above were incubated with 50 μM CuSO4-BTTAA premixed complex (CuSO4-BTTAA, molar ratio 1:2), 100 μM alkyne-Cy3, and 2.5 mM fresh sodium ascorbate for click reaction at 37 °C for 2 h. For visualizing the metabolically labeled samples, 50 μL suspensions were incubated with 50 μM CuSO4-BTTAA premixed complex (CuSO4-BTTAA, molar ratio 1:2), 100 μM alkyne-Cy5, and 2.5 mM fresh sodium ascorbate for click reaction at 37 °C for 2 h. For measuring the O-GlcNAc stoichiometry on endogenous ESRRB, 50 μL suspensions of the chemoenzymatically labeled lysates were incubated with 50 μM CuSO4-BTTAA premixed complex (CuSO4-BTTAA, molar ratio 1:2), 100 μM alkyne-PEG2KD and 2.5 mM fresh sodium ascorbate at 37 °C for 3 h, followed by protein precipitation and re-suspending the proteins with 30 μL SDS loading buffer. The samples were resolved on 10% SDS/PAGE and imaged by Typhoon FLA 9500 (GE). The gels were then stained by Coomassie Brilliant Blue (CBB) or subjected to immunoblotting to show the equal loading.

**Enrichment of O-GlcNAcylated proteins and on-bead digestion**. For identification of 1,3-Ac2GalNAz or 1,3-Pr2GalNAz modified peptides, 1 mL suspensions from metabolic labeling were incubated with 100 μM alkyne-AC-biotin, BTTAA-CuSO4 mixture (50 μM CuSO4, CuSO4-BTTAA 1:2 in molar ratio) and 2.5 mM freshly prepared sodium ascorbate at R.T. for 2 h. The resulting solution was precipitated by 10 mL methanol at −80 °C overnight. The precipitants were centrifuged (4 °C, 4200 g, 15 min), washed twice with 10 mL ice-cold methanol, and re-suspended in 1 mL PBS (pH 7.4) containing 1.2% SDS (wt/vol). Then 100 μL streptavidin beads (Thermo Fisher Scientific) were washed with 1 mL PBS (pH 7.4) for three times and re-suspended in 5 mL PBS (pH 7.4), which was transferred into

the protein solution. The resulting mixture was incubated at R.T. for 4 h with gentle rotation. The beads were then washed with PBS (pH 7.4) for five times and Milli-Q water for five times. The resulting samples were re-suspended with 500 μL 6 M urea in PBS, followed by addition of 25 μL 200 mM DTT in water (65 °C, 15 min) and 25 μL 400 mM iodoacetamide in water (35 °C, 30 min, in dark). After changing buffer with 200 μL 2 M urea in PBS, 4 μL trypsin (0.5 μg/μL in trypsin resuspension buffer) and 2 μL 100 mM CaCl2 were added and the resulting mixture was incubated at 37 °C for 16 h, followed by Acid cleavage and LC-MS/MS analysis.

For detecting ESRRB O-GlcNAcylation, 1 mL suspensions from metabolic labeling or chemoenzymatic labeling were incubated with 100 μM alkyne-PEG4-biotin, BTTAA-CuSO4 mixture (50 μM CuSO4, CuSO4-BTTAA 1:2 in molar ratio) and 2.5 mM freshly prepared sodium ascorbate at R.T. for 2 h. The resulting solution was precipitated by 10 mL methanol at −80 °C overnight. The precipitants were centrifuged (4 °C, 4200 g, 15 min), washed twice with 10 mL ice-cold methanol, and re-suspended in 1 mL mixture of resuspension buffer A [4% SDS (wt/vol) and 10 mM EDTA] and B [1% Brij 97 (wt/vol), 150 mM NaCl and 50 mM triethanolamine (TEA), pH 7.4], with the volume ratio of A:B at 1:8. To the mixture were added 100 μL streptavidin-coated agarose beads (Thermo Fisher Scientific, 20353), and the resulting mixture was incubated at R.T. for 4 h with gentle rotation. The beads were washed with the following buffer in order: 2% SDS in PBS (wt/vol, pH 7.4), 8 M urea in 250 mM ammonium bicarbonate (ABC), 2.5 M NaCl in PBS (pH 7.4), PBS (pH 7.4) for three times and Milli-Q water for three times. The agarose beads were re-suspended with 50 μL SDS loading buffer, resolved on 10% SDS/PAGE, and detected by immunoblotting.

**Acid cleavage**. The beads with modified peptides obtained above were washed with PBS (pH 7.4) for five times and Milli-Q water for five times, and cleaved twice with 200 μL 2% formic acid in water (vol/vol) for totally 2 h at R.T. with gentle rotation. The supernatant was collected and evaporated in a vacuum centrifuge, followed by LC-MS/MS analysis.

**LC-MS/MS analysis**. All MS data were acquired on an Orbitrap Elite mass spectrometer (Thermo Fisher Scientific) equipped with a 4 μm C18 bulk material (InnosepBio) packed loading column (100 μm × 2 cm), a separating C18 capillary column (75 μm × 15 cm) and an EASY-nLC 1000 system (Thermo Fisher Scientific). For mapping 1,3-Ac2GalNAz or 1,3-Pr2GalNAz modified peptides, the samples were re-constituted in 0.1% formic acid in water (vol/vol), loaded onto the pre-column and separated by the separating column eluting with 7–35% B [A: 0.1% formic acid in water (vol/vol), B: 0.1% formic acid in acetonitrile (vol/vol)] during 120 min and then subjected to ETD-based LC-MS/MS analysis. The instrument was operated with a full mass scans (300-1,700 m/z) in FT mode at a resolution of 60,000, followed by ETD MS/MS scan on the top 10 most intense precursors (with multiply charge such as 2 +, 3 +, or higher). The ETD activation times were set at 150 ms. The charge state dependent time, supplemental activation for ETD and dynamic exclusion were enabled.

**Data processing**. SwissPort *Homo sapiens* database and *Mus musculus* database were downloaded from Uniprot (www.uniprot.org) on 4 November 2016. For identification of 1,3-Ac2GalNAz or 1,3-Pr2GalNAz modified peptides, the ETD raw data files were subjected to database searches using MaxQuant[63] software suite (version 1.5.8.2) integrated with Andromeda search engine. The default parameters were set as reported[8,20]. Briefly, the modified peptides with an Andromeda score ≥ 40 and a delta score ≥8 were selected. For site identification, the spectra with a localization score ≥0.75 were considered as the high-confidence sites. Then the O-HexNAc sites located on the extracellular domains of membrane proteins or secreted proteins were excluded from the O-GlcNAc site list.

**Recombinant expression of ESRRB and OGT in *E. coli***. The expression vectors pUCBAD-MBP-OGT and pET-28a(+)-ESRRB were individually transformed or co-transformed in *E.coli* BL21 (DE3) (Transgen) and the recombination expression of OGT or/and ESRRB were induced with 0.5% arabinose (wt/vol) or/and 0.5 mM IPTG at 25 °C for 16 h in a shaker as reported[39].

**Quantitative RT-PCR Analysis**. Total RNA of mESCs R1 was extracted with the TRIzol™ Reagent (Thermo Fisher Scientific, 15596026) and 1 μg RNA was performed for reverse transcription using PrimeScript™ RT reagent Kit with gDNA Eraser (Takara, RR047A), followed by quantitative PCR analysis with PowerUp SYBR Green (Thermo Fisher Scientific). The primers used in this study are listed: *Gapdh* (the forward primer, 5′-TCACCACCATGGAGAAGGCCG-3′ and the reverse primer, 5′-TCTTCTGGGTGGCAGTGATGGC-3′), *Esrrb* (the forward primer, 5′-GGCTCGTCGGACGCCAG-3′ and the reverse primer, 5′-GTACTC GCATTTGATGGCGGAGTCC-3′), *Oct4* (the forward primer, 5′-TCCTCTGAG CCCTGTGCCGA-3′ and the reverse primer, 5′-GAGAACGCCCAGGGTGAG CC-3′), *Nanog* (the forward primer, 5′-GCGGACTGTGTTCTCTCAGGCC-3′ and the reverse primer, 5′-ACTCCACTGGTGCTGAGCCCT-3′) and *Rex1* (the forward primer, 5′-GACAAGTGGCCAGAAAGGGGCC-3′ and the reverse primer, 5′-CCGTCAGGGAAGCCATCTTCCTC-3′).

**Identification of ESRRB O-GlcNAcylation sites**. ESRRB purified from *E.coli* co-expressing OGT and ESRRB was resolved on 10% SDS/PAGE and stained by CBB. The ESRRB band was excised, washed with Milli-Q water, and destained twice with the destaining buffer (the mixture of 50 mM ABC and acetonitrile at the volume ratio of 1:1) for 30 min, followed by dehydrating in acetonitrile and rehydrating with 10 mM DTT in 50 mM ABC for 45 min at 56 °C. The gel slices were treated with 55 mM iodoacetamide in 50 mM ABC for 45 min at R.T. in the dark, followed by dehydrating in acetonitrile. Then gel slices were digested with chymotrypsin (2 ng/μL) and incubated at 37 °C for 16 h. The resulting peptides were eluted twice in 50% acetonitrile containing 5% TFA (vol/vol) with 200 μL per time and evaporated in a vacuum centrifuge. The samples were detected by an Orbitrap Elite mass spectrometer (Thermo Fisher Scientific) using the ETD fragmentation mode. The data were processed using Mascot software (version 2.3.02, MatrixScience) with a HexNAc modification of 203.079373 Da and evaluated the modification sites manually.

**FLIM-FRET imaging**. The cellular imaging of O-GlcNAcylation on a specific protein was described previously[38], and adapted for this work. Briefly, after the transfection of EGFP-ESRRB and 1 mM GalNAz treatment for 48 h, the HeLa cells were washed, fixed and permeabilized, followed by click-labeling used for FLIM-FRET imaging. FLIM-FRET imaging was performed on a TCS SP8X scanning confocal microscope (Leica). Images and distribution histograms of fluorescence lifetime were acquired by TCSPC software (SymPhoTime 64 software, PicoQuant GmbH).

**CHX treatment**. About $1 \times 10^5$ mESCs per well were plated on a 6-well, cultured overnight, pre-treated with DMSO or Ac$_4$5SGlcNAc for 48 h, and co-treated with 50 μM CHX for up to 8 h. For detecting the protein degradation of FLAG-ESRRB$^{WT}$ and FLAG-ESRRB$^{S25A}$, about $1 \times 10^5$ mESCs per well were plated on a six-well, cultured for 36 h, and incubated with 50 μM CHX for up to 12 h. For detecting the protein degradation of EGFP-ESRRB$^{WT}$ and EGFP-ESRRB$^{S25A}$, about $4 \times 10^5$ HEK293T cells per well were plated on a 6-well, cultured overnight, transfected with the respective plasmids using VigoFect based the manufacturer's instructions, followed by removing the medium and addition of fresh medium with 50 μM CHX for 12 h.

**Co-immunoprecipitation assay**. After plasmids transfection, the cells were harvested, lysed by Sucrose-NP40 buffer [250 mM sucrose, 150 mM NaCl, 5 mM MgCl$_2$, 0.5% Nonidet P-40 (vol/vol), 10 mM NaF, 1 mM DTT, 1 mM PMSF and protease inhibitor mixture (Pierce), 25 mM Tris-HCl, pH 7.5] with sonication, centrifuged at 12,000 g for 10 min at 4 °C to discard debris, and measured the protein concentration by BCA protein assay kit (Pierce). For detecting ubiquitination of ESRRB and interactions between ESRRB and NANOG or OCT4 in HEK293T cells, 1 mL 3 mg/mL cell lysates were incubated with 50 μL GFP-magnetic beads (MBL, D153-11) at 4 °C for 2 h with gentle rotation. For detecting protein interactions between OGT and ESRRB in HEK293T cells and verifying protein interactions between ESRRB and NANOG or OCT4 in mESCs, 1 mL 3 mg/mL cell lysates were incubated with 50 μL FLAG-magnetic beads (MBL, M185-11) at 4 °C for 2 h with gentle rotation. Then the beads were washed with IP wash buffer [10% Glycerol (vol/vol), 150 mM NaCl, 5 mM MgCl$_2$, 0.1% Nonidet P-40 (vol/vol), 1 mM DTT, 0.2 mM PMSF, 25 mM Tris-HCl, pH 7.5] for five times and re-suspended with 30 μL SDS loading buffer. These reacted samples were resolved on 10% SDS/PAGE and detected by immunoblotting.

**Construction of mESCs expressing ESRRB$^{WT}$ or ESRRB$^{S25A}$**. For lentivirus packing and processing, HEK293FT cells were transfected with 10 μg plasmid of pCDH-EF1-FLAG-ESRRB$^{WT}$-T2A-copGFP (or pCDH-EF1-FLAG-ESRRB$^{S25A}$-T2A-copGFP), 8 μg plasmid of PAX2, and 4 μg plasmid of MD.2G when the cells were grown to approximately 60% confluence in a 10-cm cell culture dish. The cell culture medium was filtered and collected after 72 h, followed by addition of Lenti-X$^{TM}$ Concentrator (Clontech, 631231) for lentivirus extraction based the manufacturer's instructions. The resulting sample was resolved in PBS (pH 7.4) and added into the cell culture of R1 mESCs when the cells were grown to approximately 30% confluence. After incubation for 24 h, the medium was changed to fresh R1 cell culture medium. The cells were cultured for two passages, followed by flow cytometry sorting of mESCs expressing ESRRB$^{WT}$ or ESRRB$^{S25A}$ with equal GFP fluorescence intensity.

**Luciferase assay**. About $2 \times 10^5$ HEK293T cells per well plate were seeded on a 12-well and cultured overnight. For luciferase assay in the OGT inhibition experiment, HEK293T cells co-transfecting with 1 μg ESRRB-responsive element-driven luciferase plasmids and 1 μg pCDH-EF1-FLAG-ESRRB-T2A-CopGFP were co-treated with DMSO or 50 μM Ac$_4$5SGlcNAc for 48 h, and used for luciferase activity detection. For luciferase assay in the WT and S25A ESRRB, HEK293T cells co-transfecting with 1 μg ESRRB-responsive element-driven luciferase plasmids and 1 μg pCDH-EF1-FLAG-ESRRB$^{WT}$-T2A-CopGFP (or pCDH-EF1-FLAG-ESRRB$^{S25A}$-T2A-copGFP) for 48 h were analyzed. Luciferase activity detections were performed using Luciferase Reporter Gene Assay Kit (Yeasen Biotechnology, 11401ES60).

**RNA sequencing and bioinformatics**. Total RNA was extracted using TRIzol Reagent (Thermo Scientific). Sequencing libraries were generated using NEBNext Ultra RNA Library Prep Kit for Illumina (NEB) following manufacturer's recommendations. Sequencing was performed on Illumina Hiseq X Ten. Reads from sequencing were aligned to the mouse genome using Hisat v2.1.0[64], with the index built from mm10 genome and annotation files on GENCODE. The resulting SAM alignment files were quality filtered (-q 10), converted to BAM format and sorted using SAMtools v1.3[65]. Then duplicate reads were removed by Picard tools v1.140 (http://picard.sourceforge.net). Reads aligned to exons were counted with featureCounts v1.6.3[66], and differential analysis was performed with DESeq2 v1.20.0[67] on Bioconductor.

**AP staining assay**. As reported[47], briefly, about 500 mESCs expressing Mock, ESRRB$^{WT}$ or ESRRB$^{S25A}$ were individually plated the six-well plate for 24 h, followed by removing the medium, washing twice with PBS, and addition of fresh medium with no LIF for 6 d or 9 d. When the medium turned yellow, refreshed in time. The colonies were analyzed according to the AP staining using Stemgent AP Staining Kit II (Stemgent, 00-0055).

**Teratoma formation in nude mice**. About $2 \times 10^6$ mESCs expressing ESRRB$^{WT}$ or ESRRB$^{S25A}$ were injected into 8-week immunodeficient nude mice (Vital River Laboratory Animal Center) in both flanks for two weeks. Then the mice were euthanized and the masses of teratomas were weighted. All animal experiments were approved by the Institutional Animal Care and Use Committee of Peking University.

**HE staining**. Teratomas were fixed with 4% paraformaldehyde at 4 °C for 24 h and then transferred to 30% (w/v) sucrose in PBS (pH 7.4) overnight. Tissues were embedded in O.C.T. Compound (Sakura Finetek USA Inc.) and sectioned at 10 μm on a microtome cryostat (Leica CM 1950). Tissue sections were stained with hematoxylin and eosin kit (Beyotime, C0105), and digital light microscopic images were taken on an inverted microscope (Leica DMI4000 B), equipped with a digital microscope camera (Leica DFC420 C).

**Statistics**. For comparison between two groups, the two-tailed Student's t-test was used to calculate *P* values. For comparison among multiple groups, one-way ANOVAs with the post hoc Tukey HSD Calculator were used to determine differences. Error bars represent mean ± s.d.

**Reporting summary**. Further information on research design is available in the Nature Research Reporting Summary linked to this article.

## Data availability

The raw data of RNA sequencing is available in the Gene Expression Omnibus database under the accession number GSE131246. The raw mass spectral data in our study is available via ProteomeXchange with identifier PXD015035 [http://proteomecentral.proteomexchange.org/cgi/GetDataset?ID=PXD015035]. The raw data underlying Figs. 2a–f, 3a–d, 3f–i, 4a–j, 5a, c, as well as Supplementary Figs. 2, 3, 4, 5a, 7a, b, 8, 9, 10, 14b, 16b, c, 17, 18, 20c, 21 are available in the source data file. All other data generated or analyzed during this study are available from the corresponding author on reasonable request.

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

## Acknowledgements

We thank Prof. Ye-Guang Chen at Tsinghua University for kindly providing murine R1 ESCs, Prof. Hong Wu at Peking University for the HA-UBIQUITIN plasmid, Prof. Qin Shen at Tongji University for HEK 293FT cells and the plasmids of PAX2 and MD.2G, and Prof. Wengong Yu at Ocean University for the pUCBAD-MBP-OGT plasmid. Part of the MS experiments were performed at the mass spectrometry facility of the National Center for Protein Sciences at Peking University. This project is supported by the National Natural Science Foundation of China (No. 91753206, No. 21425204, and No. 21521003), the National Key Research and Development Projects (No. 2018YFA0507600 and 2016YFA0501500).

## Author contributions

Y.H., X.F., and X.C. conceived the project; Y.H., X.F., and Y.S. performed research with the help of C.Z., D.S., K.Q., W.Q., and W.Z.; Y.H., X.F., and X.C. analyzed the data and wrote the paper.

## Additional information

**Competing interests:** The authors declare no competing interests.

**Peer Review Information:** Nature Communications thanks the anonymous reviewers for their contribution to the peer review of this work. Peer reviewer reports are available.

