## [Peer Review File · Nature Communications]

Editorial Note:

This manuscript has been previously reviewed at another journal that is not operating a transparent peer review scheme. This document only contains reviewer comments and rebuttal letters for versions considered at *Nature Communications*.

Reviewers' comments:

Reviewer #1 (Remarks to the Author):

The authors have proactively addressed almost all of my previous comments. I do believe that the paper is appropriate for publication in *Nature Communications* after the following is addressed.

1. I think I wasn't clear enough in my previous "Comment #5." I appreciate that the FLAG-tagged version of the ESRRB will not allow the overexpressed protein and the endogenous protein to be distinguished by molecular weight. I was simply suggesting to run non-transfected cells next to the overexpressing cells and blot with an ESRRB antibody. The change in intensity of the bands between the two lanes will give an idea of the levels of overexpression. This will nicely complement the RNA levels experiment that the authors did include.

Reviewer #2 (Remarks to the Author):

In this revised manuscript Hao et al. positively addressed most of the major concerns previously raised, and improved their discussion of results. In particular, the data supporting the claim that Esrrb S25A shows reduced stability and impaired interactions with Oct4 and Nanog is now sufficiently robust. The authors also improved the characterisation of the consequence of WT or mutant Esrrb overexpression, notably performing global gene expression analysis in self-renewing ES cells. Nonetheless, beyond the reported effects on protein half-life and interactions, evidence of the importance of S25 O-GlcNAcylation for the function of Esrrb as a pluripotency factor remains weak. However, we recognise that a complete functional characterisation falls beyond the scope of the current manuscript.

We reiterate the suggestion that a timecourse analysis of the kinetics of pluripotency gene downregulation during differentiation of ESCs overexpressing WT or mutant Esrrb would provide molecular information to substantiate the results of clonal assays.

Minor remarks:

The table presented in Fig S19 should be improved. It would be helpful to report gene names instead of Ensembl IDs. For instance the table shows the potentially important differential expression of DNMT3L. It would also be important to have a notion of the number of replicates on which the differential gene expression analysis is based.

On page 13 the authors state that « The auxiliary TFs of the network, such as ESRRB, KLF2, KLF4, and TBX3, are also essential for pluripotency ». These factors are important, but not essential, for pluripotency.

Reviewer #3 (Remarks to the Author):

Hao, Fan, Chen and coworkers entitled “Next-generation unnatural monosaccharides reveal that ESRRB O-GlcNAcylation enhances pluripotency of mouse embryonic stem cells” describe efforts to develop differentially esterified N-azidoacetylgalactosamine derivatives for use as metabolic probes. They use this probe to study the post-translational modification, O-GlcNAcylation, in the regulation of pluripotency in mouse embryonic stem cells. This is a revision of a work previously submitted to [redacted] which I reviewed. The authors have done an excellent job addressing all of my concerns. The new manuscript is much improved and I appreciate the extra time to do the requested experiments. The details to the chemical synthesis of the probes and biological experiments are rigorous. I support publishing this manuscript as the probes will be a helpful tool for glyco-biologists.

I have this one very minor comment:

Figure 1: Could the authors include a scale bar for the heat map?

Response to Referees Letter

Reviewer #1:

“The authors have proactively addressed almost all of my previous comments. I do believe that the paper is appropriate for publication in Nature Communications after the following is addressed.”

Response: We thank the reviewer for the positive comments. The manuscript has been revised as described below.

“I think I wasn't clear enough in my previous "Comment #5." I appreciate that the FLAG-tagged version of the ESRRB will not allow the overexpressed protein and the endogenous protein to be distinguished by molecular weight. I was simply suggesting to run non-transfected cells next to the overexpressing cells and blot with an ESRRB antibody. The change in intensity of the bands between the two lanes will give an idea of the levels of overexpression. This will nicely complement the RNA levels experiment that the authors did include.”

Response: We thank the reviewer for this suggestion. Accordingly, we have performed the suggested experiment, which has now been presented in the last paragraph of page 14 in the revised manuscript and in the supplementary Fig. 16c in the revised Supplementary Information.

Reviewer #2:

“In this revised manuscript Hao et al. positively addressed most of the major concerns previously raised, and improved their discussion of results. In particular, the data supporting the claim that Esrrb S25A shows reduced stability and impaired interactions with Oct4 and Nanog is now sufficiently robust. The authors also improved the characterisation of the consequence of WT or mutant Esrrb overexpression, notably performing global gene expression analysis in self-renewing ES cells. Nonetheless, beyond the reported effects on protein half-life and interactions, evidence of the importance of S25 O-GlcNAcylation for the function of Esrrb as a pluripotency factor remains weak. However, we recognise that a complete functional characterization falls beyond the scope of the current manuscript.”

Response: We thank the reviewer for the positive comments. We have revised the manuscript as described below.

“We reiterate the suggestion that a timecourse analysis of the kinetics of pluripotency gene downregulation during differentiation of ESCs overexpressing WT or mutant Esrrb would provide molecular information to substantiate the results of clonal assays.”

Response: We thank the reviewer for this suggestion. As suggested, we monitored the expression of pluripotency genes including *Oct4*, *Nanog* and *Rex1* during differentiation

of mESCs stably expressing S25A ESRRB or WT. The expression level of mESC marker genes including Oct4, Nanog and Rex1 decreased more rapidly in cells expressing ESRRBS25A than the WT ESRRB-expressing cells upon removing LIF from the culture medium. These data are now presented in the first paragraph of page 18 in the revised manuscript and Supplementary Fig. 21 in the revised Supplementary Information.

Minor remarks:

“The table presented in Fig S19 should be improved. It would be helpful to report gene names instead of Ensembl IDs. For instance, the table shows the potentially important differential expression of DNMT3L. It would also be important to have a notion of the number of replicates on which the differential gene expression analysis is based.”

Response: We thank the reviewer for this suggestion. We have now updated the table in Supplementary Fig. 19 as suggested.

“On page 13 the authors state that “The auxiliary TFs of the network, such as ESRRB, KLF2, KLF4, and TBX3, are also essential for pluripotency”. These factors are important, but not essential, for pluripotency.”

Response: We thank the reviewer for pointing this out. Accordingly, we have made the change as suggested.

Reviewer #3:

“Hao, Fan, Chen and coworkers entitled “Next-generation unnatural monosaccharides reveal that ESRRB O-GlcNAcylation enhances pluripotency of mouse embryonic stem cells” describe efforts to develop differentially esterified N-azidoacetylgalactosamine derivatives for use as metabolic probes. They use this probe to study the post-translational modification, O-GlcNAcylation, in the regulation of pluripotency in mouse embryonic stem cells. This is a revision of a work previously submitted to [redacted] which I reviewed. The authors have done an excellent job addressing all of my concerns. The new manuscript is much improved and I appreciate the extra time to do the requested experiments. The details to the chemical synthesis of the probes and biological experiments are rigorous. I support publishing this manuscript as the probes will be a helpful tool for glyco-biologists.”

Response: We appreciate the reviewer for the overall positive comments and recommendation for publication.

“I have this one very minor comment:

Figure 1: Could the authors include a scale bar for the heat map?”

Response: We thank the reviewer for this suggestion. We have added a scale bar as suggested.